# A novel approach for measuring allostatic load highlights differences in stress burdens due to race, sex and smoking status

Aleah Bailey[1,2], Alexis Payton[2], Jonathon Fleming[3], Julia E. Rager[1,2], Ilona Jaspers[1,2]*

**1** Curriculum in Toxicology and Environmental Medicine, UNC School of Medicine, University of North Carolina at Chapel Hill, Chapel Hill, North Carolina, United States of America, **2** Center for Environmental Medicine, Asthma, and Lung Biology, U.S. Environmental Protection Agency, University of North Carolina at Chapel Hill, Chapel Hill, North Carolina, United States of America, **3** North Carolina State University, Bioinformatics Research Center, Raleigh, North Carolina, United States of America

* ilona_jaspers@med.unc.edu

## Abstract

The psychosocial and environmental stressors resulting from adverse social determinants of health (SDOH) can result in allostatic load, or 'wear and tear' on the body due to prolonged stress. Allostatic load serves as a useful tool to measure stress- related biological responses that can be integrated into risk assessments. In this study a novel allostatic load measure was used to investigate the relationship between acute stress, stress-related physiological dysregulation and allostatic load (AL) and various demographic variables. Using a small volume of serum collected from 63 participants, we assessed stress biomarkers through ELISA assays and developed a one-sample, semi-automated method for calculating AL scores derived from the Toxicological Prioritization Index (ToxPi) framework. Our analysis employed ordinal regression models to identify the contributions of primary mediators to predicting blood pressure classification, revealing that epinephrine was the most significant predictor of blood pressure, followed by cortisol. Primarily, our results highlight racial disparities in stress loads, in which Black participants had greater secondary mediator scores, and higher AL compared to White participants. Black females and Black non-smokers had higher physiological dysregulation and allostatic load than White participants. Overall, our study presents a novel scoring approach that facilitates the integration of allostatic load measures from a single clinical sample into environmental health research. It also demonstrates the utility of allostatic load in capturing race and sex differences in stress burdens, with important implications for understanding health disparities and improving risk assessments.

## Introduction

Chronic stress and environmental health disparities stem from harmful social determinants of health (SDOH) and environmental hazards, disproportionately impacting populations based on socioeconomic, racial, and geographic factors [1]. SDOH,

**Data availability statement:** All relevant data are within the paper and its Supporting Information files.

**Funding:** AB, AP, and IJ were supported by the Environmental Protection Agency Cooperative Agreement with the University of North Carolina at Chapel Hill (UNC) CR 84033801, AB was supported by a pre-doctoral traineeship [National Research Service Award T32 ES007126] from the National Institute of Environmental Health Sciences, National Institutes of Health. This work was supported in part by a grant to UNC Chapel Hill from the Howard Hughes Medical Institute (HHMI) through the Gilliam Fellows Program.

**Competing interests:** The authors have declared that no competing interests exist.

which include the non-medical factors that affect health such as economic stability, access to education and healthcare, neighborhood quality, the built environment, and social support, play a significant role in shaping health outcomes [2]. A key mechanism through which these disparities manifest is allostatic load (AL), the cumulative physiological strain from chronic stress exposure. Accurately quantifying allostatic load is essential for understanding how social and environmental stressors contribute to potential health disparities and improving risk assessments.

Stress is often measured through self-reported questionnaires, but these are subject to bias and underreporting, especially among racial and ethnic minority groups that experience greater social stressors and generally have higher baseline allostatic load [3,4]. Biomarker-based measures provide a more objective method for assessing cumulative stress burden. Allostatic load reflects long-term physiological strain resulting from prolonged activation of the sympathetic-adrenal-medullary (SAM) and hypothalamic pituitary adrenal (HPA) axes due to chronic stress exposure [5]. Allostatic load biomarkers include primary mediators, such as cortisol, epinephrine, and noradrenaline, which directly reflect adrenal function, and secondary mediators, which capture the downstream effects of chronic stress on cardiovascular (HDL and total cholesterol), metabolic (Hba1c) and inflammatory (C-reactive protein and fibrinogen) pathways [6]. These biomarkers can then be combined into a score to provide an objective measure of cumulative stress load and its physiological consequences [7]. However, existing allostatic load scoring methods vary widely, limiting comparability across studies. This study introduces a novel, weighted scoring approach designed to improve standardization and integration of stress biomarkers identified in a single serum sample.

Despite its widespread use, allostatic load measures lack standardization [8]. Many studies classify allostatic load using high-risk quartiles based on study-specific distributions, leading to inconsistent biomarker weighting and poor cross-study comparability. To address this limitation, we propose a semi-automated scoring approach the integrates multi-dimensional biomarker data collected from a single clinical sample using scaled and weighted summation methods. Our component scoring approach applies principles from the Toxicological Prioritization Index (ToxPi) framework, which has been used for chemical risk assessment and environmental health studies [9–15]. By deriving component weightings using semi-automated methods, our model provides a systematic, reproducible approach to calculate allostatic load that enhances comparability across studies. Additionally, we compare allostatic load with acute stress markers to illustrate how chronic stress burden may be underestimated when relying on short-term physiological responses alone. This study introduces a novel allostatic load scoring approach, demonstrating its potential as a biomarker-based stress assessment that overcomes key limitations in existing measurement methods.

## Materials and methods

### Participants

Study participants consisted of healthy adults aged 18–43 years who were 1) non-smokers not routinely exposed to secondhand tobacco smoke or 2) active

cigarette smokers. Participants provided self-reported demographic information, including sex, race and ethnicity, as well as information on medical history. Participants with a history of asthma, chronic obstructive pulmonary disease (COPD), any immunodeficiency or any cardiorespiratory condition were excluded from the study. Active cigarette smoking was determined by measuring levels of serum cotinine, the metabolite of nicotine that serves as a biomarker of daily nicotine use. A final sample size of 63 participants were included in this analysis. Since this is a retrospective analysis integrating data from four studies, blood pressure measurements were not collected in one study (IRB #05–2547), resulting in systolic and diastolic blood pressure data available for 34 of the 63 participants, which was used to develop the acute stress score. Informed consent was obtained for all participants. Participants were recruited in 2009 and 2013–2016 by protocols approved by the Institutional Review Board at the University of North Carolina at Chapel Hill (IRB #: 05–2547, 13–2246, 13–3076, 13–3454).

### Serum collection

Whole blood samples were collected in BD Vacutainer serum-separating tubes (Fisher Scientific, Waltham, MA) and allowed to clot for a minimum of 15 minutes at room temperature. Blood samples were centrifuged at 1200 x g for 10 minutes, and the serum layer was transferred to a fresh tube and stored at -80°C until samples analyses.

### Stress biomarker analysis

Protein expression of stress biomarkers was measured in serum. All mediators were quantified using specific commercial enzyme-linked immunosorbent assays (ELISA) according to manufacturer's instructions: Noradrenaline, Epinephrine, hemoglobin A1c (HbA1c) (LSBio, Seattle, Washington), Cortisol (R&D Systems, Minneapolis, MN), C-reactive protein (CRP) (Meso Scale Diagnostics, Rockville, MD), high-density lipoprotein (HDL) and Fibrinogen (Abcam, Cambridge, United Kingdom). 96-well plates were either pre-coated or coated overnight at room temperature with capture antibodies, as instructed by the manufacturer. Absorbance was measured at 450nm using a CLARIOstar® Plus microplate reader (Cary, NC) or Meso Sector S 600MM.

### Stress biomarker data processing

Allostatic load biomarkers were subsequently processed, and the succeeding analyses were carried out in R software (v 4.3.1.). Initially, background filters were implemented to remove AL biomarkers with less than 25% of the data in the entire dataset and all variables passed this filter. With 17.5% of its data missing, epinephrine was the only variable with missing data. This was addressed by performing data imputation using the Quantile Regression Imputation of Left-Censored data (QRILC) using the *imputeLCMD* package [16]. QRILC was selected to impute missing data, since the missing biomarker values were likely attributable to low expression or missing not at random (MNAR) and QRILC generates data from the left side of a normal distribution.

### Quantification of stress using physiological indicators

**Mediator score calculation.** A dimensionless score was first generated for each allostatic load biomarker based on a min-max normalization, akin to the ToxPi framework. This constrains the value of each mediator to fall between 0 and 1 as shown in Formula (1):

$$Mediator\ Score_{HDL} = \frac{HDL_{sample} - HDL_{overall\ min}}{HDL_{overall\ max} - HDL_{overall\ min}} \quad (1)$$

(1)

**Acute stress score summation.** Primary mediators, cortisol, epinephrine and noradrenaline, reflect acute stress and were used to develop an acute stress score [17]. Weights for three primary mediators were derived using ordinal regression modeling, as described in Fleming et al. [18]. First, mediator scores were pseudo $\log_2$ transformed to make the

data more normally distributed. Next, we performed ordinal regression modeling that predicted blood pressure categorized into either two or three classes. For the three-blood pressure class model, participants were grouped into normal (systolic BP < 120 mm Hg and diastolic BP < 80 mm Hg, n = 15), elevated (systolic BP 120–129 mm Hg and diastolic BP < 80 mm Hg, n = 4), and hypertensive stage 1 (systolic BP 130–139 mm Hg or diastolic BP 80–89 mm Hg, n = 11) and stage 2 (systolic BP ≥ 140 mm Hg or diastolic BP ≥ 90 mm Hg, n = 4), with a total of 15 hypertensive participants [19]. For the two-blood pressure class model, participants were grouped into normal (n = 15) or elevated and hypertensive stage 1 and 2 (n = 19). Ordinal regression predicts an ordinal outcome with at least two classes using continuous data. Here, ordinal regression leveraged the primary mediator scores to predict blood pressure classification. The goal of this analysis was not to obtain the predictions of a participant's blood pressure class, but rather the weights or the relative importance attributed to each of the three primary mediators when predicting blood pressure class. The weights were then extracted from both ordinal regression models resulting in two sets of weights, which were rescaled so that their sum added up to 1. As a result, the weight of each mediator fell between 0 and 1. Each rescaled weight was multiplied by its respective mediator score and summed to generate acute stress scores for each participant as shown in Formula (2).

$$Acute\ Stress\ Score = \sum Weight_{primary\ mediator} \times Mediator\ Score_{primary\ mediator} \tag{2}$$

**Secondary mediator score summation.** Secondary mediators (HDL, HbA1c, Fibrinogen, CRP) representing the cardiometabolic and immune effects of stress, were summed according to Formula (3). Similar to the primary mediators, secondary mediator scores were pseudo $\log_2$ transformed. Since secondary mediators lacked a physiological proxy to use in the ordinal regression model for deriving weights, each mediator was assigned a weight of 0.25, the average of the mediators, resulting in values that ranged from 0 to 1. Additionally, given the inverse relationship with stress and atherosclerosis, HDL was assigned a weight of -0.25 [20,21].

$$Secondary\ Mediator\ Score = \sum 0.25 \times Mediator\ Scores_{secondary\ mediators} - 0.25 \times Mediator\ Score_{HDL} \tag{3}$$

**Allostatic Load Score Summation.** Allostatic load scores were calculated as the sum of acute stress and secondary mediator scores, as shown in Formula (4), and resulted in values between 0 and 2.

$$Allostatic\ Load\ Score = Acute\ Stress\ Score + Chronic\ Stress\ Score \tag{4}$$

### Stress distribution analysis

T-tests were used to evaluate mean differences in acute stress, secondary mediators, and allostatic load across three groups, race, sex and smoking status. A p-value < 0.05 was considered significant. Additionally, Two-way ANOVA tests were used to evaluate mean differences in acute stress, secondary mediators, and allostatic load in groups stratified by sex, race, or smoking status. This analysis compared participants within multiple strata including sex and race, sex and smoking status, and race and smoking status. A p-value < 0.05 was considered significant. Pairwise t-tests were subsequently run and resulting p values were adjusted ($P_{adj}$) for multiple tests using false discovery rate (FDR) $q$ values. Similarly, $P_{adj} < 0.05$ was considered significant. Boxplots showing the overall and stratified comparisons were visualized using the *ggpubr* package [22].

## Results

### Participant demographics

The mean age of the participants was 28 years old, ranging from 18 to 43 years. Around 63% of the participants were women and about 49% were non-Hispanic Black (Table 1). Of the participants with blood pressure measurements (n = 34), 15 had normal blood pressure, 4 had elevated blood pressure, 11 had stage 1 hypertension, and 4 had stage 2 hypertension.

**Table 1. Demographic characteristics, stress biomarker concentrations and stress scores of study participants used for stress scoring analysis.**

| | Male (n = 23) | | | | Female (n = 40) | | | |
|---|---|---|---|---|---|---|---|---|
| | White (n = 12) | | Black (n = 11) | | White (n = 20) | | Black (n = 20) | |
| | NS (n = 4) | CS (n = 8) | NS (n = 3) | CS (n = 8) | NS (n = 12) | CS (n = 8) | NS (n = 11) | CS (n = 9) |
| **Age** (Mean ± SD) | 23.5 ± 4.64 | 28 ± 5.1 | 33.3 ± 9.5 | 29 ± 5.9 | 25.8 ± 4.4 | 25.8 ± 7.3 | 30.2 ± 6.6 | 32.4 ± 4.2 |
| **Primary Mediators** | | | | | | | | |
| **Cortisol** (ng/mL) | 58.6 ± 45.4 | 92.8 ± 41.6 | 94.3 ± 17.4 | 96.1 ± 43.3 | 123.0 ± 77.7 | 84.6 ± 52.8 | 117.6 ± 71.7 | 73.8 ± 28.7 |
| **Epinephrine** (ng/mL) | 7.9 ± 4.3 | 6.5 ± 2.0 | 7.2 ± 2.1 | 5.9 ± 1.8 | 6.7 ± 1.4 | 5.2 ± 0.84 | 6.9 ± 1.5 | 5.8 ± 1.3 |
| **Norepinephrine** (ng/mL) | 2.41 ± 1.47 | 2.83 ± 1.77 | 1.73 ± 0.60 | 3.07 ± 1.46 | 2.60 ± 1.82 | 1.04 ± 0.63 | 2.20 ± 1.53 | 1.86 ± 1.76 |
| **Secondary Mediators** | | | | | | | | |
| **Fibrinogen** (µg/mL) | 1.18 ± 0.55 | 1.01 ± 0.54 | 0.86 ± 0.14 | 1.32 ± 0.79 | 1.08 ± 0.83 | 0.80 ± 0.69 | 1.19 ± 0.75 | 0.77 ± 0.31 |
| **CRP** (µg/mL) | 9.52 ± 15.21 | 3.5 ± 5.14 | 3.99 ± 5.71 | 1.73 ± 1.36 | 4.42 ± 8.26 | 7.38 ± 12.40 | 10.13 ± 14.08 | 6.45 ± 9.42 |
| **Hba1c** (mg/mL) | 12.95 ± 10.03 | 12.3 ± 8.96 | 10.03 ± 7.49 | 13.32 ± 10.85 | 7.31 ± 7.10 | 4.70 ± 7.24 | 9.89 ± 8.26 | 8.04 ± 7.84 |
| **HDL** (mg/mL) | 1.68 ± 0.58 | 1.7 ± 0.84 | 1.80 ± 1.08 | 2.24 ± 0.86 | 1.96 ± 0.74 | 2.62 ± 2.15 | 1.83 ± 0.67 | 1.87 ± 1.56 |
| **Acute Stress Two** | 0.33 ± 0.24 | 0.33 ± 0.11 | 0.29 ± 0.12 | 0.24 ± 0.08 | 0.32 ± 0.18 | 0.24 ± 0.12 | 0.34 ± 0.09 | 0.23 ± 0.09 |
| **Acute Stress Three** | 0.39 ± 0.29 | 0.30 ± 0.14 | 0.25 ± 0.16 | 0.19 ± 0.06 | 0.26 ± 0.14 | 0.18 ± 0.09 | 0.31 ± 0.11 | 0.21 ± 0.11 |
| **Secondary Mediator Sores** | 0.37 ± 0.27 | 0.42 ± 0.17 | 0.39 ± 0.21 | 0.37 ± 0.22 | 0.30 ± 0.22 | 0.28 ± 0.27 | 0.63 ± 0.27 | 0.46 ± 0.17 |
| **Allostatic Load Two** | 0.70 ± 0.27 | 0.72 ± 0.22 | 0.68 ± 0.34 | 0.57 ± 0.21 | 0.63 ± 0.33 | 0.46 ± 0.26 | 0.98 ± 0.31 | 0.67 ± 0.20 |
| **Allostatic Load Three** | 0.76 ± 0.25 | 0.75 ± 0.21 | 0.64 ± 0.37 | 0.62 ± 0.22 | 0.56 ± 0.31 | 0.52 ± 0.27 | 0.94 ± 0.32 | 0.69 ± 0.21 |

Data represented as mean ± standard deviation

## Ordinal regression model weights

In the ordinal regression model with three blood pressure categories (normal, elevated, hypertensive), epinephrine was the strongest predictor (73%), followed by cortisol (22%) and noradrenaline (4%) (Fig 1A). In the two-class model (normal vs elevated/hypertensive), epinephrine's contribution decreased to 53%, while cortisol's increased to 45% (Fig 1B). These results align with studies linking both epinephrine and cortisol to blood pressure regulation [23]. Scores generated with weights from the three-class model demonstrated greater significance and were applied in subsequent analyses and figures (table in S4 Table).

## Stress distribution analysis

T-tests and ANOVA were employed to assess significant differences in acute stress, secondary mediator, and allostatic load scores due to race, sex, or smoking status (data in S4-S6 Tables). Non-smokers had greater acute stress (p = 0.035) and higher allostatic load, albeit not statistically significant (p = 0.089), compared to smokers (Fig 2C and I). This may be driven by the elevated levels of epinephrine in non-smokers compared to smokers (p = 0.054) (data in S3 Table and S3 Fig). ANOVA tests showed no significant differences among groups in acute stress scores (Fig 3A–C).

Black participants had higher secondary mediator scores than White participants ($P_{adj}$ = 0.009) (Fig 2D). Specifically, Black females had greater secondary mediator scores than White females ($P_{adj}$ = 0.003) and Black non-smokers had greater secondary mediator scores than both White non-smokers ($P_{adj}$ = 0.018) and smokers ($P_{adj}$ = 0.026), indicative of greater physiological dysregulation (Fig 3D and E). Black participants also had greater AL than white participants, albeit not statistically significant ($P_{adj}$ = 0.068) (Fig 2G). Black females had higher AL scores than White females ($P_{adj}$ = 0.006) and higher scores than Black males, albeit not statistically significant ($P_{adj}$ = 0.081). Similarly, White males had higher AL than White females, albeit not statistically significant ($P_{adj}$ = 0.081), suggesting that the relationship between sex and stress burdens may vary by race (Fig 3G). Black non-smokers had the highest overall AL ($P_{adj}$ = 0.028), suggesting that race plays a significant role in physiological stress burdens, particularly among non-smoking Black individuals (Fig 3H).

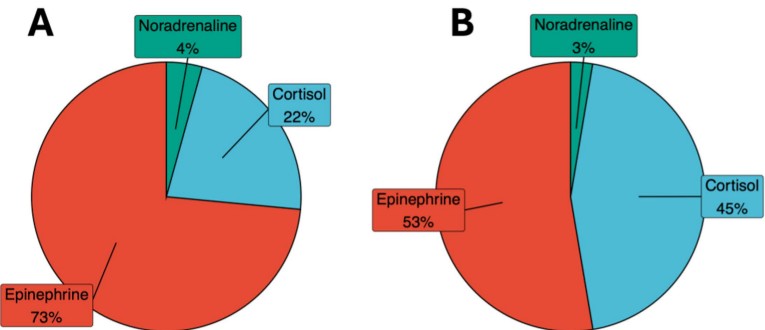

**Fig 1. Biomarkers' contribution to ordinal regression models.** Three acute stress biomarkers (epinephrine, cortisol, and noradrenaline) were used to predict a participant's blood pressure category. Blood pressure was categorized into 3 classes: normal, at risk, and hypertensive and (B) 2 classes: normal and at risk and hypertensive participants combined. Each mediator's contribution to the model was extracted and depicted as a percentage in the pie chart.

## Discussion

This study examines trends in acute stress, secondary mediators, and allostatic load scores using three scoring approaches, including a novel, regression-based partially weighted method for assessing allostatic load. Our novel scoring approach enhances reproducibility by assigning data-driven weights to primary mediators using regression-based modeling. Furthermore, it enables comprehensive stress assessment from a single clinical sample, increasing its practical utility. Ordinal regression analysis demonstrated that epinephrine was the most significant predictor of blood pressure classification among the three primary mediators. Epinephrine has direct effects on blood pressure through binding to adrenergic receptors on smooth muscle cells of blood vessels, which causes vasoconstriction and vascular resistance, leading to elevated blood pressure [24]. Additionally, several studies have described the mechanistic link between epinephrine and hypertension [25,26].

In our study, non-smokers had higher epinephrine than smokers, which may underlie the observation of greater acute stress and allostatic load scores in non-smokers compared to smokers. Nicotine is known to activate the HPA axis and increase levels of cortisol, noradrenaline and epinephrine. However, habitual smokers may exhibit blunted stress responses compared to non-smokers [27–29]. These inconsistencies regarding the relationship between smoking and stress responses are likely due to the variability in sample populations (smoking history and dependence), differences in stress measurements and timing, and length of exposure to nicotine [29].

There were no significant differences in acute stress due to race or sex. Acute stress activates the sympathetic adrenal medullary (SAM) and hypothalamic pituitary adrenal (HPA) axis, depending on appraisal of the stressor, resulting in the release of stress hormones which modify various physiological systems as the body attempts to subdue the stress response. Stress appraisals differentiate a challenge, or significant events with high controllability, from a threat, situations with high personal significance, poor controllability and limited resources for coping [30,31]. Although racial and ethnic minorities face greater social stress burdens, acute stress responses were not significantly different by race. This may stem from differences in stress appraisals, as well as the transient and context-dependent nature of acute stress markers [3,30].

However, our secondary mediator and allostatic load measures, which detected racial and sex differences in stress burdens, exemplify the need for cumulative stress assessments. When analyzing differences in secondary mediators between groups, a measure of cardiometabolic and immune dysregulation due to stress, we observed Black participants had higher scores than White participants, which were largely driven by Black females and non-smokers. These trends are even more pronounced when analyzing the combination of acute and secondary mediators, summed into the allostatic load score. Our results suggest that Black participants exhibited higher allostatic load scores. These findings illustrate

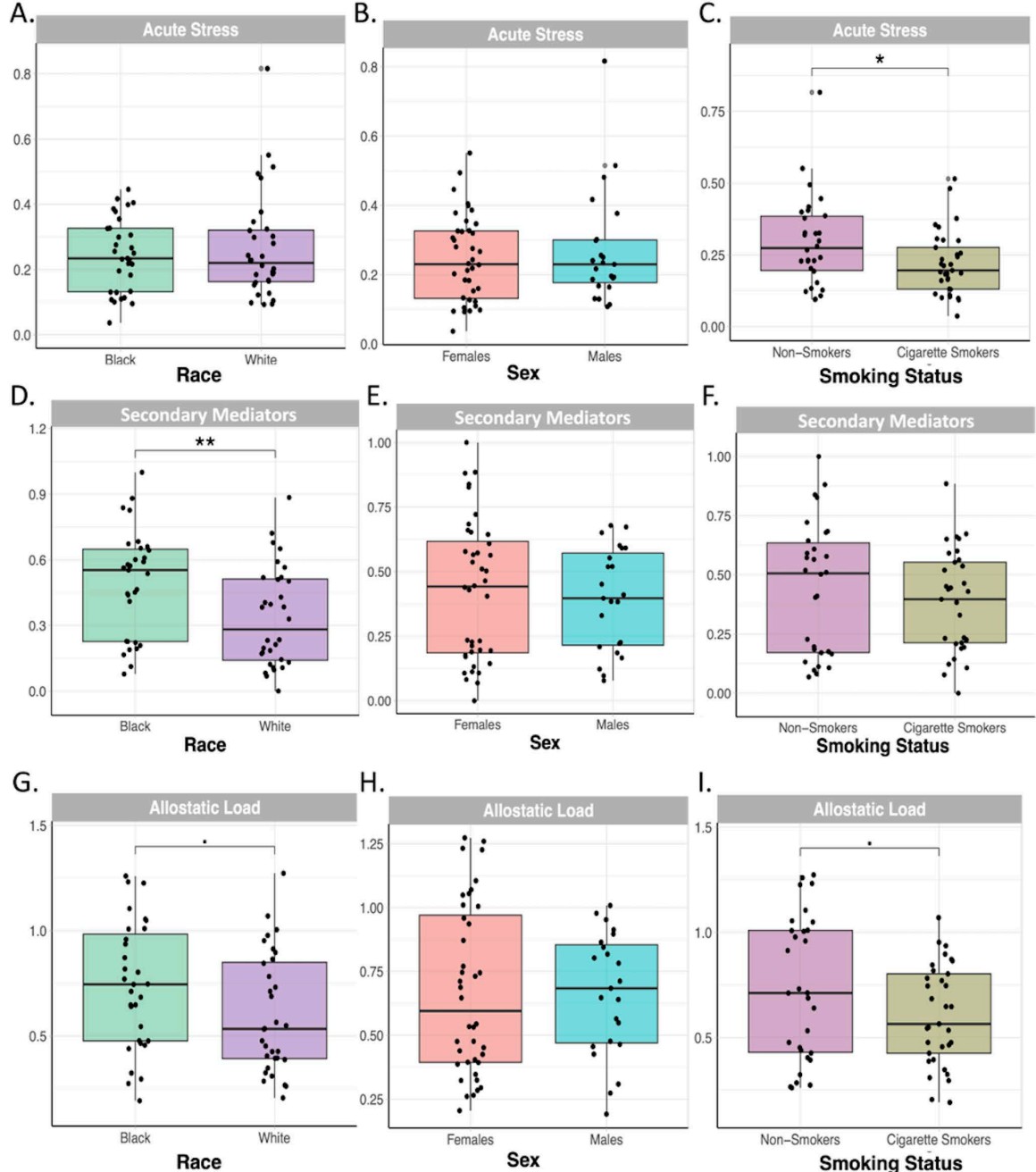

**Fig 2. Stress outcomes by race, sex and smoking status.** Stress outcomes measured include (A-C) acute stress, (D-F) secondary mediator scores and (G-I) allostatic load. Groups were compared using T-tests. Significant differences are denoted as follows: ▪ Represents significance between groups (p<0.1), * represents significance between groups (p<0.05), ** represents significance between groups (p<0.01), *** represents significance between groups (p<0.001).

how our scoring method can highlight potential disparities in chronic stress burdens. Additionally, we observed an inverse relationship between sex and allostatic load among White and Black participants suggesting that the interaction between stress and sex varies by race.

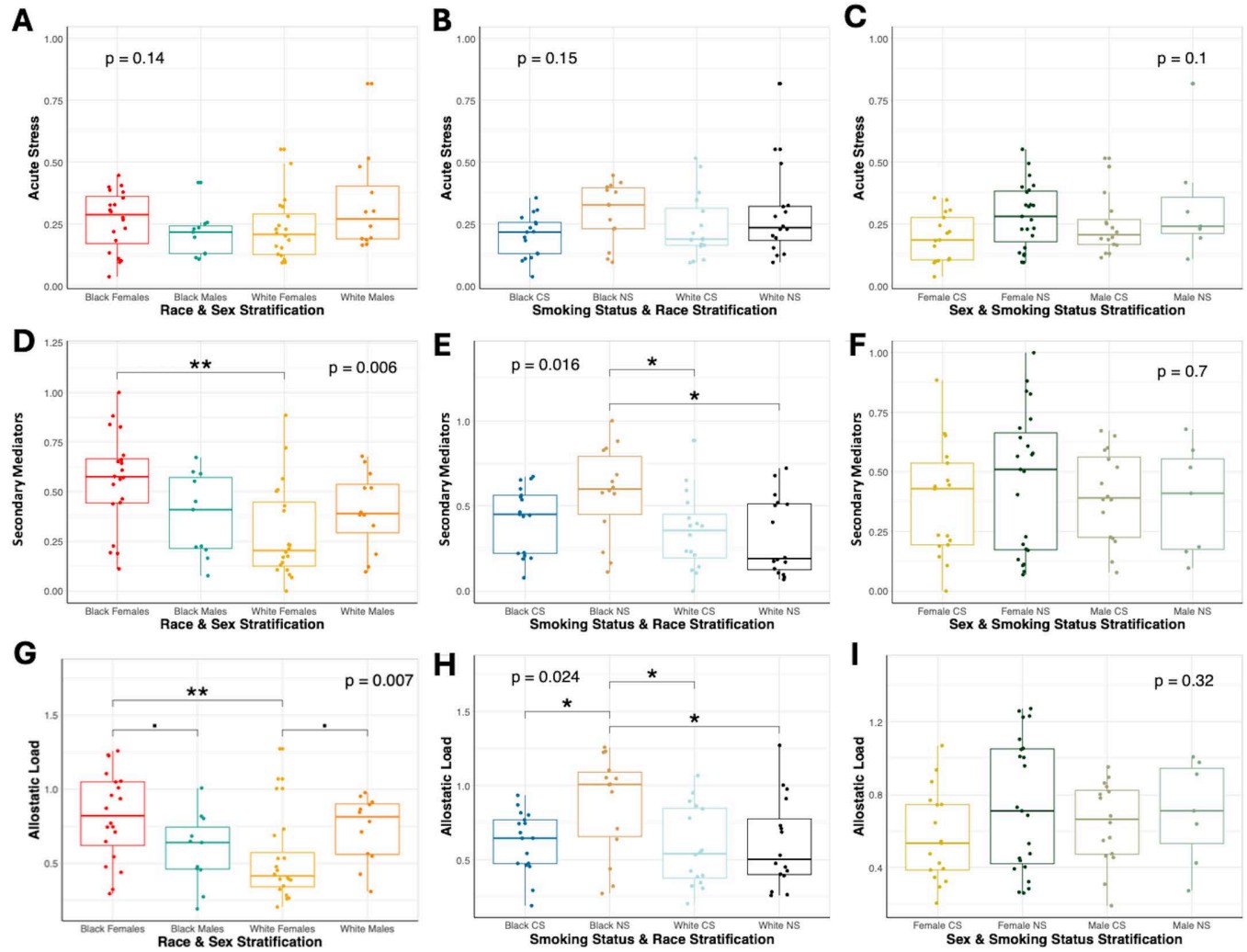

**Fig 3. Stratified stress outcomes by race, sex and smoking status.** Stress outcomes measured include (A-C) acute stress, (D-F) secondary mediator scores and (G-I) allostatic load. Groups were compared using a two-way ANOVA test to obtain an overall p-value as depicted in the upper left- or right-hand corner of each plot. Pairwise t-tests were then run for post-hoc testing and include the following comparisons: ▪ Represents significance between groups ($P_{adj} < 0.1$), * represents significance between groups ($P_{adj} < 0.05$), ** represents significance between groups ($P_{adj} < 0.01$), *** represents significance between groups ($P_{adj} < 0.001$).

The observed racial and sex disparities in stress burdens align with the "weathering hypothesis," which suggests that chronic exposure to adverse SDOH contributes to early health deterioration and disparities [32–36]. This disproportionate stress burden may explain why Black women, regardless of socioeconomic position, often exhibit the highest allostatic load [34,37,38]. According to this hypothesis, 'weathering' may accelerate physiological dysfunction, increase chronic disease risk, and contribute to higher mortality among Black individuals [32,35]. However, the disproportionate risk of allostatic load for Black women has been hypothesized to result, in part, from the compounded burden of navigating both gender inequalities and race- and class- based discrimination. In response to stress, Black women are believed to adopt maladaptive coping strategies including suppressing emotions, working harder to prove worthiness, excessive self-reliance, overcommitting to help others, and postponing self-care, which could further contribute to physiological stress burdens [39]. These findings may have implications for maternal-child health, as chronic stress has been associated with

adverse birth outcomes. Further, this may have broader implications on clinical and research study inclusion. The SWAN study, for example, examined women's health during menopause; however, the early exclusion of Black women limited its insights into the unique heath challenges and stress-related disparities they face, impacting the study's overall relevance [40]. Thus, addressing these disparities is essential for developing more inclusive health research and effective interventions tailored for racial and ethnic minority groups.

While our study offers valuable insights into the distinctions between acute stress and allostatic load measures in capturing race and sex disparities, several limitations should be noted. The use of serum-based physiological markers is both a strength and limitation of the study. In clinical settings, stress biomarkers are often measured across a range of samples, including plasma, saliva or urine. However, since our focus is on relative allostatic load, our findings demonstrate that small serum volumes can effectively measure a range of stress biomarkers, allowing for relative allostatic load assessments within a cohort using a single collected blood sample. Another limitation is the moderate sample size and sensitivity of acute stress markers, which may have resulted in our study not being sufficiently powered to detect significant differences by sex and race, despite clear differences in allostatic load. Future studies that include larger and more diverse cohorts, controlled conditions, and behavioral assessments of stressors and coping mechanisms may be needed to better capture acute stress responses and validate these findings. Acute stress is a short-term response measured in our study through cortisol, noradrenaline and epinephrine, whereas allostatic load reflects cumulative physiological strain from chronic stress exposure. Acute stress responses can be influenced by coping mechanisms that buffer physiological reactivity, even in individuals with high allostatic load, as may be the case in our study, where smokers exhibited significantly lower acute stress than non-smokers. This may also occur when the perceived severity of a stressor does not align with its physiological impact, which may account for the lack of discernable trends in acute stress due to race. In contrast, allostatic load, which integrates multiple physiologic systems, provides a more stable measure of chronic stress burden. Additionally, although the use of secondary mediators is customary for allostatic load assessments, it is understood that they may oversimplify the complex physiological responses induced by chronic stress. This limitation, however, does not detract from the value of the study's novel allostatic load measure, which integrates weighted primary mediators and offers a more comprehensive assessment.

## Conclusion

In summary, this study demonstrates the importance of assessing cumulative stress and AL to capture physiological burden beyond acute stress markers. Using a modified allostatic load scoring approach, we observed differences in acute stress and allostatic load between smokers and non-smokers, suggesting potential differences in stress response regulation. Additionally, our findings indicate that Black participants had higher physiological dysregulation and allostatic load scores compared to White participants, with trends largely driven by Black women. This new scoring approach described here presents a reproducible and adaptable allostatic load scoring method, enhancing its applicability in risk assessments and stress-related health research, which can be easily applied in future research without requiring multi-sample collection. Additionally, it provides flexibility and adaptability allowing the inclusion of additional allostatic load biomarkers such as dehydroepiandrosterone sulfate (DHEA-s), a HPA axis antagonist that indicates adrenal gland function, or body mass index (BMI) an indicator of obesity [7]. By enabling comprehensive stress evaluation from a single sample, our approach provides a foundation for further validation in larger epidemiological and clinical cohorts to establish its broader applicability.

Future directions should include validating this approach in larger, more diverse cohorts to refine its utility in evaluating how stress influences biological responses to environmental exposures and contributes to health disparities.

## Supporting information

**S1 Table. T-test results for Individual Biomarker analysis between males and females.** For each biomarker, the table includes the calculated t-statistic, degrees of freedom (df), p-value. Significance levels are denoted as follows: ▪Represents significance between group (P < 0.1), * Represents significance between groups (p < 0.05), ** represents

significance between groups (p<0.01), \*\*\* represents significance between groups (p<0.001), \*\*\*\* represents signifi-cance between groups (p<0.0001).
(DOCX)

**S1 Fig. Boxplots for t-test results for Individual Biomarker analysis between males and females.** Significance lev-els are denoted as follows: ▪Represents significance between group (P<0.1), \* Represents significance between groups (p<0.05), \*\* represents significance between groups (p<0.01), \*\*\* represents significance between groups (p<0.001), \*\*\*\* represents significance between groups (p<0.0001).
(DOCX)

**Table S2. T-test results for Individual Biomarker analysis between Black and White participants.** For each bio-marker, the table includes the calculated t-statistic, degrees of freedom (df), p-value. Significance levels are denoted as follows: ▪Represents significance between group (P<0.1), \* Represents significance between groups (p<0.05), \*\* rep-resents significance between groups (p<0.01), \*\*\* represents significance between groups (p<0.001), \*\*\*\* represents significance between groups (p<0.0001).
(DOCX)

**S2 Fig. Boxplots for t-test results for Individual Biomarker analysis between Black and White participants.** Signifi-cance levels are denoted as follows: ▪Represents significance between group (P<0.1), \* Represents significance between groups (p<0.05), \*\* represents significance between groups (p<0.01), \*\*\* represents significance between groups (p<0.001), \*\*\*\* represents significance between groups (p<0.0001).
(DOCX)

**S3 Table. T-test results for Individual Biomarker analysis between smokers and non-smokers.** For each biomarker, the table includes the calculated t-statistic, degrees of freedom (df), p-value. Significance levels are denoted as follows: ▪Represents significance between group (P<0.1), \* Represents significance between groups (p<0.05), \*\* represents sig-nificance between groups (p<0.01), \*\*\* represents significance between groups (p<0.001), \*\*\*\* represents significance between groups (p<0.0001).
(DOCX)

**S3 Fig. Boxplots for t-test results for Individual Biomarker analysis between smokers and non-smokers.** Signifi-cance levels are denoted as follows: ▪Represents significance between group (P<0.1), \* Represents significance between groups (p<0.05), \*\* represents significance between groups (p<0.01), \*\*\* represents significance between groups (p<0.001), \*\*\*\* represents significance between groups (p<0.0001).
(DOCX)

**S4 Table. T-test results for stress scores compared across groups (race, sex and smoking status).** Results are shown for acute and allostatic load scores calculated from weights derived from both the two- and three- blood pressure classes input into the ordinal regression model. P-values<0.1 were considered significant. ▪ Represents significance between groups (p<0.1), \* represents significance between groups (p<0.05), \*\* represents significance between groups (p<0.01), \*\*\* represents significance between groups (p<0.001).
(DOCX)

**S5 Table. Two-way ANOVA data comparing groups by race, sex and smoking status.** ▪Represents significance between group (P<0.1), \* Represents significance between groups (p<0.05), \*\* represents significance between groups (p<0.01), \*\*\* represents significance between groups (p<0.001), \*\*\*\* represents significance between groups (p<0.0001).
(DOCX)

**S6 Table. Pairwise t-test post-hoc test results assessing differences in stratified groups.** ▪Represents significance between group ($P_{adj} < 0.1$), * Represents significance between groups ($P_{adj} < 0.05$), ** represents significance between groups ($P_{adj} < 0.01$), *** represents significance between groups ($P_{adj} < 0.001$), **** represents significance between groups ($P_{adj} < 0.0001$).
(DOCX)

**S7 Table. Raw data for allostatic load biomarkers.** The table includes serum biomarker concentrations measured using ELISA and associated demographic information.
(DOCX)

**S8 Table. Raw data for acute, secondary and allostatic load scores derived from serum biomarkers.** Scores were calculated using weighted formulas based on two- and three-class ordinal regression models. Secondary mediators did not have a physiological proxy to calculate weighted scores and therefore only have a single score reported.
(DOCX)

## Acknowledgments

The authors acknowledge and thank the study coordinators for their dedication an effort in recruiting participants, facilitating sample collection and processing, and managing study operations. The authors also thank the study participants for their time and commitment.

## Author contributions

**Conceptualization:** Aleah Bailey, Julia E Rager, Ilona Jaspers.

**Data curation:** Aleah Bailey, Alexis Payton.

**Formal analysis:** Aleah Bailey, Alexis Payton, Jonathon Fleming, Julia E Rager.

**Funding acquisition:** Ilona Jaspers.

**Investigation:** Aleah Bailey, Ilona Jaspers.

**Methodology:** Aleah Bailey, Alexis Payton, Jonathon Fleming, Julia E Rager.

**Project administration:** Ilona Jaspers.

**Resources:** Ilona Jaspers.

**Supervision:** Julia E Rager.

**Validation:** Alexis Payton.

**Writing – original draft:** Aleah Bailey, Alexis Payton, Jonathon Fleming, Ilona Jaspers.

**Writing – review & editing:** Aleah Bailey, Alexis Payton, Jonathon Fleming, Julia E Rager, Ilona Jaspers.

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
