## [Decision Letter · Decision Letter 0]

11 Mar 2025

PONE-D-24-53214A Novel Approach for Measuring Allostatic Load Highlights Differences in Stress Burdens due to Race, Sex and Smoking StatusPLOS ONE

Dear Dr. Jaspers,

Thank you for submitting your manuscript to PLOS ONE. After careful consideration, we feel that it has merit but does not fully meet PLOS ONE’s publication criteria as it currently stands. Therefore, we invite you to submit a revised version of the manuscript that addresses the points raised during the review process.

We look forward to receiving your revised manuscript.

Kind regards,

katsuya oi, PhD

Academic Editor

PLOS ONE

“AB, AP, and IJ were supported by the Environmental Protection Agency Cooperative Agreement with the University of North Carolina at Chapel Hill (UNC) CR 84033801, AB was supported by a pre-doctoral traineeship [National Research Service Award T32 ES007126] from the National Institute of Environmental Health Sciences, National Institutes of Health. This work was supported in part by a grant to UNC Chapel Hill from the Howard Hughes Medical Institute (HHMI) through the Gilliam Fellows Program.”

“AB, AP, and IJ were supported by the Environmental Protection Agency Cooperative Agreement with the University of North Carolina at Chapel Hill (UNC) CR 84033801, AB was supported by a pre-doctoral traineeship [National Research Service Award T32 ES007126] from the National Institute of Environmental Health Sciences, National Institutes of Health. This work was supported in part by a grant to UNC Chapel Hill from the Howard Hughes Medical Institute (HHMI) through the Gilliam Fellows Program.”

“AB, AP, and IJ were supported by the Environmental Protection Agency Cooperative Agreement with the University of North Carolina at Chapel Hill (UNC) CR 84033801, AB was supported by a pre-doctoral traineeship [National Research Service Award T32 ES007126] from the National Institute of Environmental Health Sciences, National Institutes of Health. This work was supported in part by a grant to UNC Chapel Hill from the Howard Hughes Medical Institute (HHMI) through the Gilliam Fellows Program.”

Reviewers' comments:

Reviewer's Responses to Questions

**Comments to the Author**

1. Is the manuscript technically sound, and do the data support the conclusions?

Reviewer #1: Yes

Reviewer #2: Yes

Reviewer #3: No

2. Has the statistical analysis been performed appropriately and rigorously? 

Reviewer #1: Yes

Reviewer #2: Yes

Reviewer #3: I Don't Know

3. Have the authors made all data underlying the findings in their manuscript fully available?

Reviewer #1: Yes

Reviewer #2: Yes

Reviewer #3: Yes

4. Is the manuscript presented in an intelligible fashion and written in standard English?

Reviewer #1: Yes

Reviewer #2: Yes

Reviewer #3: Yes

5. Review Comments to the Author

Reviewer #1: Page 4; Lines 84-86: “Systolic and diastolic blood pressure data was available for 34 participants and used to develop the acute stress score.” Where do you explain why only 34 of 63 participants had blood pressure data? Perhaps I missed that, seems here would be a good spot.

Page 4; Lines 93-97: “Blood containing tubes were then centrifuged and serum was stored at -80°C. The blood was then centrifuged at 1200 x g for 10 minutes, and the serum layer was transferred to a fresh tube and stored at -80°C until samples were collected from all participants.” Repetitious? Why a semi-repeated sentence?

Page 7; line 162: “…demographic groups, race, sex and smoking status.” Never have read of smoking status being categorized as a demographic group before. Interesting categorization.

Page 17; Lines 396-397: Why is a Table inserted into References List?

Overall, an interesting method for assessing allostatic load.

Reviewer #2: Dear Author,

Thank you so much for the opportunity to review this article. I think with a few revisions, it will be ready to go! Here are some recommendations:

-Introduction:

-Line 49: - I understand your need to state this, but it is very abrupt here. I recommend moving it down to line 57 and combining it here to highlight its importance in relation to measurable biomarkers as the indicators.

-Methods:

-2.5.2- Line 130: Can you please identify in the first ordinal regression model what systolic/diastolic blood pressures you are utilizing to categorize the three-class model (normal, elevated, and hypertensive stage 1 and 2) vs two class model (normal, elevated)? I understand this is just being used to describe why you weighted your data the way you did. But, it’s useful to readers who may have notions on what is elevated vs HTN 1 or 2.

-2.6-2.7-Line 156-162: Can you please describe why you used a p-value of <.1 for your t-tests and ANOVAs, when regular statistical significance is usually <.05?

-Results:

-3.1-Line 175: Please identify “The mean age of participants was 28 years old, ranging from XX years to XX years.”

-3.1- Line 175-176: Were these self-reported genders and/or race/ethnicities or did they come from a medical record? Please identify either here or in section 2.1 when identifying who your participants are and how you collected their demographic data; this is especially important to explain when working with marginalized groups and you have a pretty substantial non-Hispanic Black subgroup (49%). Especially because you talk about this in the Discussion.

-Discussion:

-No significant differences in acute stress due to race or sex- this is interesting and should be addressed in the Limitations, as it is kind of an outlier compared to prior literature.

-References: Please double check your last reference as it is running below your Table 1.

Thank you very much for this opportunity.

Jenny

Reviewer #3: Thank you for the opportunity to review your manuscript! Exploring allostatic load measurement using the Toxicological Prioritization Index (ToxPi) framework is a novel and exciting approach. However, this manuscript fails to exhibit a nuanced understanding of allostatic load as a preclinical indicator, its evolving history in measurement, and the existing knowledge regarding racial disparities of allostatic load that have been studied for decades.

The manuscript lacks a declared purpose/objective, and that lack of clear focus diminishes it. If invited by the editor for a revise and resubmit, I highly recommend narrowing the scope of the paper to address exclusively the novel method you propose, proving the concept, and use your background section to exhibit deep understanding of how allostatic load measurement has evolved and the gap in current measurement that you are addressing. I strongly recommend dropping any inferences from your novel proof of concept intended to be generalized for an entire population, especially with a small sample size. Group differences tests cannot illustrate causation. Some of the causal inferences you ground in your review of extant literature are not justified.

Prominent examples include:

“Adverse psychosocial experiences and SDOH can lead to chronic activation of the hypothalamic pituitary adrenal (HPA) axis, resulting in allostatic load, or stress-related physiological dysregulation” (line 51-52; McEwen, 2022). I believe this statement is more accurate than not; however, allostatic load has likely had temporal variations over one’s lifecourse, and this statement is more true for childhood and adolescence. There are better sources to cite than a brother’s commentary on his sibling’s life work in a memorial article about another scientific field. Why not cite the source article where possible causation was explored in scientific specificity?

McEwen C.A., McEwen B.S. Social structure, adversity, toxic stress, and intergenerational poverty: an early childhood model. Annu. Rev. Sociol. 2017;43:445–472. doi: 10.1146/annurev-soc-060116-053252.

“However, the disproportionate risk of allostatic load for Black women is likely due to the added burden of navigating gender inequalities in addition to race- and class- based discrimination, and adopting maladaptive behaviors to cope such as suppressing emotions, working harder to prove worthiness, excessive self reliance, overcommitting to help others, and postponing self-care” (line 248-251; Allen et al, 2019). The cited study explicitly states that it was intended to be a hypothesis-generating study and that it should not be interpreted in any way as inferring causation.

6. PLOS authors have the option to publish the peer review history of their article (what does this mean? ). If published, this will include your full peer review and any attached files.

**Do you want your identity to be public for this peer review?** For information about this choice, including consent withdrawal, please see our Privacy Policy .

Reviewer #1: No

Reviewer #2: **Yes: ** Jenny Clift

Reviewer #3: No

---

## [Author Response · Author response to Decision Letter 0]

11 Apr 2025

A novel approach for measuring allostatic load highlights differences in stress burdens due to race, sex and smoking status

Manuscript ID: PONE-D-24-53214 Response to the Reviewers

Editorial revisions:

1. The Financial Disclosure statement was modified as following and moved from the Acknowledgements section to the Funding Statement section of the online submission form as stated in the cover letter: “AB, AP, and IJ were supported by the Environmental Protection Agency Cooperative Agreement with the University of North Carolina at Chapel Hill (UNC) CR 84033801, AB was supported by a pre-doctoral traineeship [National Research Service Award T32 ES007126] from the National Institute of Environmental Health Sciences, National Institutes of Health. This work was supported in part by a grant to UNC Chapel Hill from the Howard Hughes Medical Institute (HHMI) through the Gilliam Fellows Program. The funders had no role in study design, data collection and analysis, decision to publish, or preparation of the manuscript.”

2. We have modified the data availability statement to indicate that all data are included within the main manuscript or Supporting Information files. Specifically, Table S7 provides raw data for allostatic load biomarker concentrations, and Table S8 contains the raw stress score data

3. Captions for our Supporting Information files have now been included at the end of the manuscript. Additionally, in-text citations have been modified to align with the Supporting Information guidelines.

4. We have modified out acknowledgements statement as stated in the cover letter: “The authors acknowledge and thank the study coordinators for their dedication an effort in recruiting participants, facilitating sample collection and processing, and managing study operations. The authors also thank the study participants for their time and commitment.”

5. We have revised the manuscript to comply with PLOS ONE’s style requirements and updated the file names to align with the journal’s naming guidelines.

Reviewer #1

Comment: Page 4; Lines 84-86: “Systolic and diastolic blood pressure data was available for 34 participants and used to develop the acute stress score.” Where do you explain why only 34 of 63 participants had blood pressure data? Perhaps I missed that, seems here would be a good spot.

Response: We apologize for the confusion. This is a retrospective study of data collected from four different studies. However, one of the studies (IRB # 05-2547) did not collect blood pressure data during study visits. To ensure this is explicitly stated in the methods section, an additional sentence was included stating “Since this is a retrospective analysis integrating data from four studies, blood pressure measurements were not collected in one study (IRB #05-2547), resulting in systolic and diastolic blood pressure data available for 34 of the 63 participants, which was used to develop the acute stress score.” (Lines 95-97).

Comment: Page 4; Lines 93-97: “Blood containing tubes were then centrifuged and serum was stored at -80°C. The blood was then centrifuged at 1200 x g for 10 minutes, and the serum layer was transferred to a fresh tube and stored at -80°C until samples were collected from all participants.” Repetitious? Why a semi-repeated sentence?

Response: We apologize for this oversight and thank the reviewer for pointing this out. We acknowledge the redundancy in the original text and have revised it for clarity and conciseness. The updated sentence now reads: “Blood samples were centrifuged at 1200 x g for 10 minutes, and the serum layer was transferred to a fresh tube and stored at -80°C until samples analyses.” (Lines 103-105).

Comment: Page 7; line 162: “…demographic groups, race, sex and smoking status.” Never have read of smoking status being categorized as a demographic group before. Interesting categorization.

Response: We recognize that smoking status is typically considered a dynamic, behavioral, or health-related variable rather than a static demographic characteristic. To improve clarity and ensure appropriate categorization, we have revised the sentence to distinguish smoking status from demographic factors. The updated sentence now reads, “Additionally, Two-way ANOVA tests were used to evaluate mean differences in acute stress, secondary mediators, and allostatic load in groups stratified by sex, race, or smoking status.” (Lines 173 – 174). Additionally, to ensure this change is consistent throughout the manuscript, we have removed “demographic” from sentences in lines 171 and 210. Finally, Figure legends were updated to “Fig 2. Stress outcomes by race, sex and smoking status” (Line 205) and “Fig 3. Stratified stress outcomes by race, sex and smoking status” in (Line 210).

Comment: Page 17; Lines 396-397: Why is a Table inserted into References List?

Overall, an interesting method for assessing allostatic load.

Response: Thank you for your careful review of the manuscript and feedback. The table was accidentally placed within the references list due to a formatting error. We have now corrected this issue by repositioning the table appropriately and included it directly after the first paragraph in which it is cited page 9 (Lines 182 – 184).

Reviewer #2

Comment: Line 49: - I understand your need to state this, but it is very abrupt here. I recommend moving it down to line 57 and combining it here to highlight its importance in relation to measurable biomarkers as the indicators.

Response: We thank the reviewer for their suggestion. We have restructured that paragraph to read: “Allostatic load reflects long-term physiological strain resulting from prolonged activation of the sympathetic-adrenal-medullary (SAM) and hypothalamic pituitary adrenal (HPA) axes due to chronic stress exposure [5]. Allostatic load biomarkers include primary mediators, such as cortisol, epinephrine, and noradrenaline, which directly reflect adrenal function, and secondary mediators, which capture the downstream effects of chronic stress on cardiovascular (HDL and total cholesterol), metabolic (Hba1c) and inflammatory (C-reactive protein and fibrinogen) pathways [6]. These biomarkers can then be combined into a score to provide an objective measure of cumulative stress load and its physiological consequences [7].” (Lines 61 – 68). This revision improves the flow and better connects allostatic load to how it is measured with biomarkers.

Comment: Methods:-2.5.2- Line 130: Can you please identify in the first ordinal regression model what systolic/diastolic blood pressures you are utilizing to categorize the three-class model (normal, elevated, and hypertensive stage 1 and 2) vs two class model (normal, elevated)? I understand this is just being used to describe why you weighted your data the way you did. But, it’s useful to readers who may have notions on what is elevated vs HTN 1 or 2.

Response: To clarify, we used the standard American Heart Association (AHA) guidelines for blood pressure classification to categorize participants into the three-class and two-class models. To explicitly state this in the manuscript, we have revised the sentence to read: “For the three-blood pressure class model, participants were grouped into normal (systolic BP < 120 mm Hg and diastolic BP < 80 mm Hg, n=15), elevated (systolic BP 120 – 129 mm Hg and diastolic BP < 80 mm Hg, n=4), and hypertensive stage 1 (systolic BP 130 – 139 mm Hg or diastolic BP 80 - 89 mm Hg, n=11) and stage 2 (systolic BP � 140 mm Hg or diastolic BP � 90 mm Hg, n=4), with a total of 15 hypertensive participants” (Lines 137 – 141).

Comment: Methods: -2.6-2.7-Line 156-162: Can you please describe why you used a p-value of <.1 for your t-tests and ANOVAs, when regular statistical significance is usually <.05?

Response: We apologize for this confusion and have revised the phrasing in the manuscript to clarify that the significance threshold is set to p<0.05 for all analyses. The p< 0.1 values are presented to highlight trends, while p<0.05 remains the threshold for statistical significance. Additionally, we have updated the results to ensure that this distinction is clear:

• Line 200-202 has been modified to “Non-smokers had greater acute stress (p=0.035) and higher allostatic load, albeit not statistically significant (p=0.089), compared to smokers (Fig 2C and 2I).”

• Line 220-221 has been modified to “Black participants also had greater AL than white participants, albeit not statistically significant (p=0.068) (Fig 2G).”

• Lines 221-223 has been modified to “Black females had higher AL scores than White females (p=0.006) and higher scores than Black males, albeit not statistically significant (p=0.081). Similarly, White males had higher AL than White females, albeit not statistically significant (p=0.081), suggesting that the relationship between sex and stress burdens may vary by race (Fig 3G)”.

Comment: Results:-3.1-Line 175: Please identify “The mean age of participants was 28 years old, ranging from XX years to XX years.”

Response: The sentence has been updated to “Study participants consisted of healthy adults aged 18-43 years who were 1) non-smokers not routinely exposed to secondhand tobacco smoke or 2) active cigarette smokers.” (Line 87-88)

Comment: Results-3.1- Line 175-176: Were these self-reported genders and/or race/ethnicities or did they come from a medical record? Please identify either here or in section 2.1 when identifying who your participants are and how you collected their demographic data; this is especially important to explain when working with marginalized groups and you have a pretty substantial non-Hispanic Black subgroup (49%). Especially because you talk about this in the Discussion.

Response: We thank the reviewer for this suggestion. To clarify, sex and race/ethnicity were self-reported by the participants. To explicitly state this in the methods section, we have added the following sentence to Section 2.1, “Participants provided self-reported demographic information, including sex, race and ethnicity, as well as information on medical history.” (Line 88-89)

Comment: Discussion- No significant differences in acute stress due to race or sex- this is interesting and should be addressed in the Limitations, as it is kind of an outlier compared to prior literature.

Response: We thank the reviewer for their feedback. The limitations section has been expanded to include a discussion on the limitations of not detecting differences in acute stress due to sex and race. (Lines 282 – 304)

Comment: References: Please double check your last reference as it is running below your Table 1.

Response: We apologize for this mistake – the Table has been moved within the manuscript file (Lines 182-184).

Reviewer #3:

Comment: The manuscript lacks a declared purpose/objective, and that lack of clear focus diminishes it. If invited by the editor for a revise and resubmit, I highly recommend narrowing the scope of the paper to address exclusively the novel method you propose, proving the concept, and use your background section to exhibit deep understanding of how allostatic load measurement has evolved and the gap in current measurement that you are addressing. I strongly recommend dropping any inferences from your novel proof of concept intended to be generalized for an entire population, especially with a small sample size. Group differences tests cannot illustrate causation. Some of the causal inferences you ground in your review of extant literature are not justified.

Response: We appreciate the reviewer’s feedback and acknowledge the need for a clearer statement of purpose. The study aims to introduce and validate a novel allostatic load measure, emphasizing its contribution to the field rather than using it to make broad inferences about population-level differences. We have significantly revised the introduction (Lines 50-84) to clarify the overall objective and the discussion to better contextualize our results and avoid causal language and overgeneralizations. Specifically, our revised Introduction focuses on a) the need for unbiased biomarker assessment of allostatic load and b) the limitations of current allostatic load approaches. Additionally, we have expanded the limitations section to mention the moderate sample size of our study.

Comment: Prominent examples include: “Adverse psychosocial experiences and SDOH can lead to chronic activation of the hypothalamic pituitary adrenal (HPA) axis, resulting in allostatic load, or stress-related physiological dysregulation” (line 51-52; McEwen, 2022). I believe this statement is more accurate than not; however, allostatic load has likely had temporal variations over one’s life course, and this statement is more true for childhood and adolescence. There are better sources to cite than a brother’s commentary on his sibling’s life work in a memorial article about another scientific field. Why not cite the source article where possible causation was explored in scientific specificity? (McEwen C.A., McEwen B.S. Social structure, adversity, toxic stress, and intergenerational poverty: an early childhood model. Annu. Rev. Sociol. 2017;43:445–472. doi: 10.1146/annurev-soc-060116-053252.)

Response: We agree that allostatic load varies over the life course. To address the reviewer’s concern, we revised the statement to “Biomarker-based measures provide a more objective method for assessing cumulative stress burden. Allostatic load reflects long-term physiological strain resulting from prolonged activation of the sympathetic-adrenal-medullary (SAM) and hypothalamic pituitary adrenal (HPA) axes due to chronic stress exposure” (Lines 60-63). Additionally, we appreciate the reviewer’s note regarding the citation and have corrected this mistake by referencing the source article discussing mechanisms underlying the relationship between stress and allostatic load (McEwen & Stellar, 1993), Line 50.

Comment: “However, the disproportionate risk of allostatic load for Black women is likely due to the added burden of navigating gender inequalities in addition to race- and class- based discrimination and adopting maladaptive behaviors to cope such as suppressing emotions, working harder to prove worthiness, excessive self-reliance, overcommitting to help others, and postponing self-care” (line 248-251; Allen et al, 2019). The cited study explicitly states that it was intended to be a hypothesis-generating study and that it should not be interpreted in any way as inferring causation.

Response: We apologize for this oversight and acknowledge that Allen et al., 2019 is a hypothesis-generating study, and have revised the sentence to ensure it does not imply causation. The sentence now reads, “However, the disproportionate risk of allostatic load for Black women has been hypothesized to result, in part, from the compounded burden of navigating both gender inequalities and race- and class- based discrimination. In response to stress, Black women are believed to adopt maladaptive coping strategies including suppressing emotions, working harder to prove worthiness, excessive self-reliance, overcommitting to help others, and postponing self-care, which could further contribute to physiological stress burdens (39)” (Lines 272-275). We believe it is important to include some discussion of potential contributing factors to the observed higher allostatic load scores in Black women and believe this article provides valuable insights; however, acknowledge that it was not proven but hypothesized in the cited article. The revised wording maintains the mention of this article while making it clear that these findings are exploratory.

---

## [Editor Report · Decision Letter 1]

15 Apr 2025

A novel approach for measuring allostatic load highlights differences in stress burdens due to race, sex and smoking Status

PONE-D-24-53214R1

Dear Dr. Jaspers,

We’re pleased to inform you that your manuscript has been judged scientifically suitable for publication and will be formally accepted for publication once it meets all outstanding technical requirements. Per my independent reading of the revised manuscript, I applaud the authors' detailed attention to the feedback they received, including valuable insights from Reviewer #3. I believe the revisions are appropriate and well executed. 

Kind regards,

katsuya oi, PhD

Academic Editor

PLOS ONE
---

## [Editor Report · Acceptance letter]

PONE-D-24-53214R1

PLOS ONE

Dear Dr. Jaspers,

I'm pleased to inform you that your manuscript has been deemed suitable for publication in PLOS ONE. Congratulations! Your manuscript is now being handed over to our production team.

Kind regards,

on behalf of

Dr. katsuya oi

Academic Editor

PLOS ONE